# Microrobots powered by concentration polarization electrophoresis (CPEP)

Florian Katzmeier [1] & Friedrich C. Simmel [1] ✉

Second-order electrokinetic flow around colloidal particles caused by concentration polarization electro-osmosis (CPEO) can result in a phoretic motion of asymmetric particle dimers in a homogeneous AC electrical field, which we refer to as concentration polarization electro-phoresis (CPEP). To demonstrate this actuation mechanism, we created particle dimers from micron-sized silica spheres with sizes 1.0 $\mu$m and 2.1 $\mu$m by connecting them with DNA linker molecules. The dimers can be steered along arbitrarily chosen paths within a 2D plane by controlling the orientation of the AC electric field in a fluidic chamber with the joystick of a gamepad. Further utilizing induced dipole-dipole interactions, we demonstrate that particle dimers can be used to controllably pick up monomeric particles and release them at any desired position, and also to assemble several particles into groups. Systematic experiments exploring the dependence of the dimer migration speed on the electric field strength, frequency, and buffer composition align with the theoretical framework of CPEO and provide parameter ranges for the operation of our microrobots. Furthermore, experiments with a variety of asymmetric particles, such as fragmented ceramic, borosilicate glass, acrylic glass, agarose gel, and ground coffee particles, as well as yeast cells, demonstrate that CPEP is a generic phenomenon that can be expected for all charged dielectric particles.

According to Smoluchowski's century-old theory, electrophoresis of colloidal particles is shape-independent[1]. In combination with the time reversibility of hydrodynamics at low Reynolds numbers, shape-independence implies that even asymmetric particles will not display any net movement when subjected to a homogeneous AC electric field. However, under experimental conditions which generate nonlinear electrokinetic phenomena, particles with a broken symmetry can experience directed propulsion also in homogeneous AC electric fields. Such AC electrophoretic propulsion was first theoretically proposed[2] for strongly polarizable particles based on induced charge electroosmosis (ICEO)[3] and later experimentally verified for metallo-dielectric Janus particles, which were observed to move perpendicular to the electric field direction[4].

In this work, we investigate a novel propulsion mechanism for weakly polarizable particles with a non-zero surface charge based on the phenomenon of concentration polarization electroosmosis (CPEO). CPEO was recently theoretically described[5] and experimentally validated[5–8] and is found to produce similar flow patterns around spheres in an AC electric field as ICEO, but under different experimental conditions. We therefore expected that similar to propulsion via induced charge electrophoresis (ICEP) resulting from ICEO, asymmetric particles subjected to CPEO would also experience directed propulsion, which we accordingly refer to as concentration polarization electrophoresis (CPEP).

In the most widely utilized experimental setup, microswimmers are placed on an electrode and exposed to a vertical electric field. Within this setup, it was demonstrated that asymmetric colloidal dimers[9] and metallo-dielectric Janus particles[10,11] are propelled perpendicularly to the electric field in a random direction in the 2D plane. To introduce maneuverability, magnetic fields have been used in

[1]Department of Bioscience, TUM School of Natural Sciences, Technical University Munich, D-85748 Garching, Germany. ✉e-mail: simmel@tum.de

combination with ferromagnetic metallo-dielectric Janus particles[12] and ferromagnetic asymmetric colloidal dimers[13]. Further, it has been demonstrated that metallo-dielectric Janus particles can be used to transport other dielectric particles[12,14,15]. In the case of asymmetric colloidal dimers, the propulsion mechanism is based on the electro-hydrodynamic interplay between electrode and particles[16,17].

Since, in contrast to this propulsion mechanism, CPEO does not require an electrode in close proximity, we surmised that it could be applied to propel asymmetric colloidal dimer particles using an in-plane electric field. Taking advantage of electro-orientation[18–21], which orients prolate particles parallel to an AC electric field through induced dipole alignment and induced hydrodynamic flows, we thus expected to achieve directed propulsion of asymmetric dimers along the field lines rather than perpendicular to them.

In the following, we demonstrate that asymmetric dimer 'microrobots' can be precisely maneuvered using a straightforward electrical setup without any additional magnetic forces by simply controlling the orientation of a homogeneous AC electric field in the plane of movement. Such AC electrically-controlled 2D actuation was previously only achieved through dielectrophoresis[22], which requires electric field gradients and a computer-controlled feedback mechanism[19,20]. We also develop a strategy to pick up, transport, and release spherical cargo particles with these microrobots by making use of induced dipole-dipole interactions and hydrodynamic flow fields. Next, we explore the dependence of the microrobots' migration speeds on electric field strength, frequency, and buffer composition, finding that these measurements align reasonably well with the theoretical predictions of CPEO. Lastly, we argue that most particles with broken symmetry can be propelled either through ICEP for metal particles or CPEP for dielectric particles. This leads us to conclude that propulsion in an AC-electric field, namely AC-electrophoresis (ACEP), is a universal phenomenon anticipated for most asymmetric particle types. We validate this by observing directed migration of a variety of asymmetric particles within a homogeneous AC-electric field. These include fragmented ceramic, borosilicate glass, acrylic glass, agarose gel, and ground coffee particles, as well as yeast cells.

## Results

### Asymmetric colloidal microswimmers in an AC electrical field

It is known that the axisymmetric fluid flow depicted in Fig. 1b arises around weakly polarizable particles with a non-zero surface charge when subjected to an AC electrical field in a low-ionic strength aqueous medium. Fluid flows towards the particle in the direction of the electric field and is repelled perpendicularly to the electric field[5,7]. We expected that for an asymmetric dimeric particle an asymmetric flow would arise as proposed in Fig. 1c that would lead to the propulsion of the particle. We experimentally verified the proposed structure of the flow field using tracer particles (Supplementary Information 3.2, Supplementary Fig. 5). Further, a dimeric particle will also align with the external electric field as shown in Fig. 1c due to an alignment torque caused by the induced dipole and fluid flow. In combination with the propulsion this leads to a directed motion along the field lines of the electric field. The movement of the dimers can thus be easily controlled by changing the direction and strength of the external AC electric field.

### Experimental setup and fabrication of particle dimers

For our experiments we designed the sample chamber shown in Fig. 1a, in which two microfluidic channels equipped with platinum electrode pairs at their inlets intersect in the center. The electric field in the center of the chamber is a superposition of the fields generated by the remote electrode pairs. Hence, the field at the intersection is homogeneous, and its direction and amplitude can be controlled by applying different electric field strengths to the channels[23,24].

We created two electric signals with the sound card of a computer, which were amplified in two stages using custom-built amplifiers before feeding them into the microchannels. With our setup we can apply AC voltages with an amplitude of up to 305 V which corresponds to an electric field amplitude of approximately 60 mV/$\mu$m in the center of our chamber. We programmed a python script to control the amplitude of the electric signals via the XY-deflection of the analog joystick of a gamepad (an Xbox Controller) which is conventionally used to play video games. As a result, the direction and amplitude of the AC electric field in our sample chamber and thereby the movement of our microrobots can be directly controlled with a joystick while imaging them with a microscope. We also included the possibility to change the field frequency to predefined values 250 Hz and 750 Hz by pressing the buttons available on the controller. Images of the setup together with detailed information on its design and manufacture are given in Supplementary Figs. 1 and 2.

Asymmetric particle dimers acting as microrobots were synthesized through the self-assembly of two differently sized, DNA-coated silica spheres with diameters 1.01 $\mu$m and 2.12 $\mu$m, respectively[25–28]. To

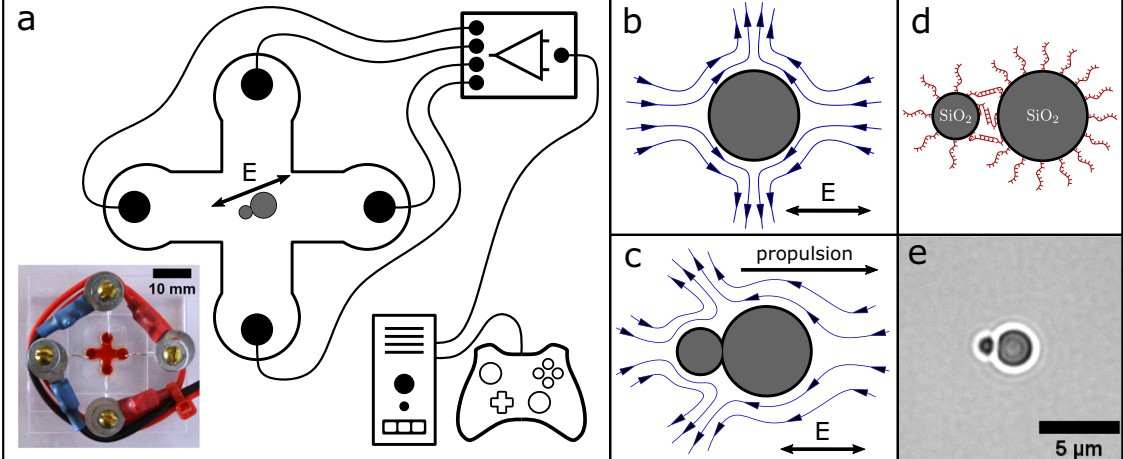

**Fig. 1 | Experimental setup and propulsion mechanism. a** Schematic representation and photograph of our experimental setup that enables control of the direction and amplitude of an AC electric field in a microscopy chamber. A dimer is drawn in the center of the cross-shaped fluidic chamber, which aligns with the externally applied AC field through an induced dipole. For visualization, the fluidic chamber in the photograph is filled with a red dye. **b** Electrokinetic flow around a spherical particle arising in an AC electric field. **c** Expected electrokinetic flow around an asymmetric particle dimer in an AC electric field, which results in directed propulsion. **d** DNA modified colloids form a dimer through DNA hybridization. **e** Microscopy image of a particle dimer.

this end, each particle type was modified with 60 nt long single-stranded DNA molecules, which had 30 nt long sub-sequences that were complementary to sequences on the respective other particle type. When mixed in the presence of 4 mM MgCl$_2$, the silica spheres specifically bound to each other via DNA duplex formation (cf. Fig. 1d & e). For our experiments, we diluted the dimers in Tris buffer (100 μM, pH 8.4) supplemented with 5.2 μM MgCl$_2$. A detailed description of synthesis and sample preparation is given in the Methods.

### Movement and maneuverability of the microrobots

Our protocol for the assembly of the silica particles resulted in a mixture of mainly monomers and dimers with only small amounts of higher order multimers. Upon exposure to an AC electric field in our sample chamber, the dimers are subject to a torque due to an induced dipole and fluid flow which aligns the dimer axis parallel to the electric field lines. The dimers can assume two alternative, stable orientations in the AC field, in which the positions of the larger and smaller particle are exchanged with each other (Fig. 2a). Notably, the induced asymmetric hydrodynamic flow around each dimer propels them in the direction defined by the position of the larger particle. Thus the particle dimers shown in the scheme of Fig. 2a would be expected to move in opposite directions, as indicated by the blue pointers. To demonstrate this effect in the experiment, we recorded a microscopy video of two differently aligned dimers while slowly changing the direction of the applied electric field using the joystick. As expected, the dimers were observed to move anti-synchronously, meaning that the trajectory of one particle was the point reflection of the other (the image

sequence shown in Fig. 2a is the first part of Supplementary Movie 1; details on video processing are given in the Methods section). All dimers in a sample move along the electric field lines collectively, with the larger particles in the front. We thus focused on the movement of individual dimers in all further experiments.

To demonstrate microrobot maneuverability, we recorded a microscopy video, in which we steered a microrobot along a slalom course around islands of monomeric particles, which remained stationary in the AC field (cf. Fig. 2b and second part of Supplementary Movie 1). The monomeric particles appear as clouds in the overlay image, since they are subject to Brownian motion. We also found a slight drift in our microscopy videos due to bulk fluid motion which we corrected by tracking several of the stationary monomeric particles and shifting the recorded video by their average displacement.

We also recorded a microscopy video, in which we maneuvered a microrobot along a racetrack adopted from a computer game. For this purpose, we printed the racetrack on a cling film and attached it with tape to the screen of the computer controlling the microscope to enable visual feedback and control by a human operator. Fig. 2c shows an overlay of the racetrack and video images recorded during the experiment (cf. third part of Supplementary Movie 1).

### Pick-up, transport, and release of cargo particles

We found that the microrobots can be readily used to pick up, transport, and release other, monomeric cargo particles, for which we utilized both electric and hydrodynamic interactions between microrobot and cargo. Fig. 3a shows a microscopy image of a

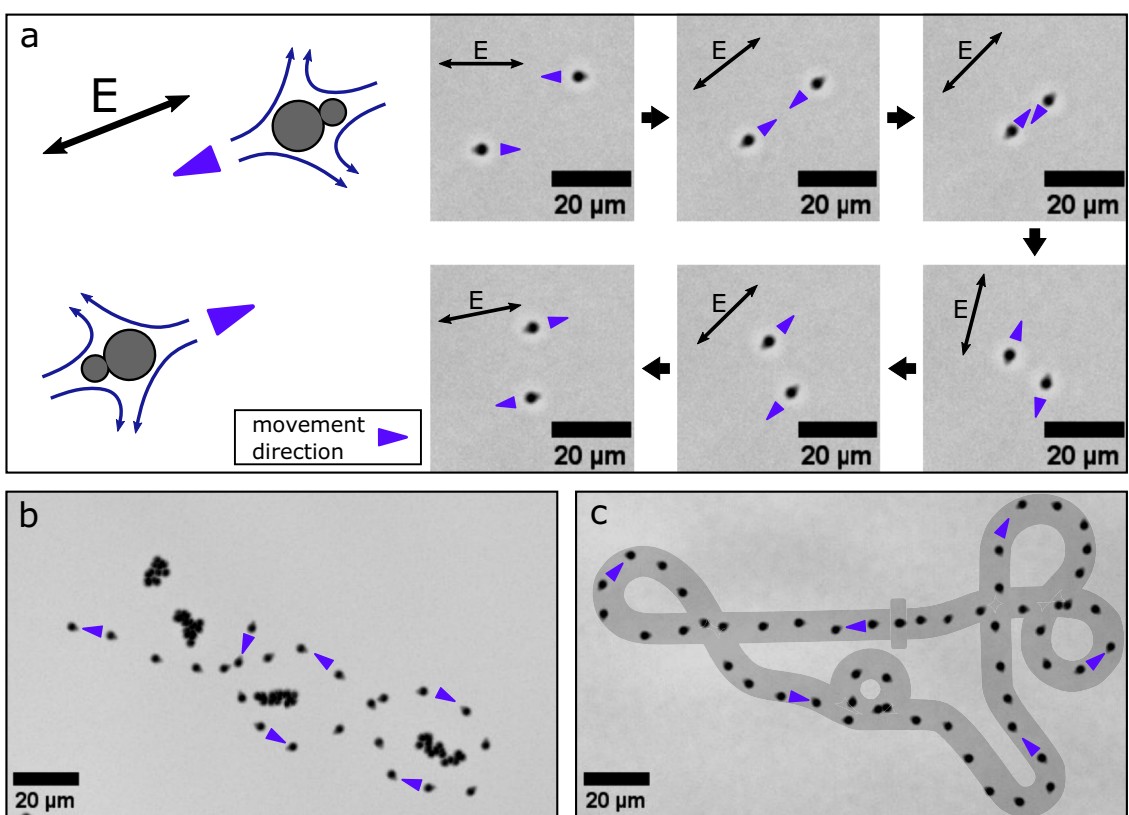

**Fig. 2 | Controlled movement of particle dimers. a** The sketch on the left shows two microrobots aligned with an AC electric field. The orientation of the AC electric field is indicated with a black double arrow. The movement direction is indicated with a blue pointer and is opposite for the two microrobots due to their opposite orientations. The image sequence on the right shows successive frames of a microscopy video demonstrating the resulting anti-synchronous movement in an electric field with slowly changing orientation. **b** By applying joystick-controlled AC voltages a microrobot is maneuvered through a slalom course around stationary monomer particles. The image shows an overlay of successive frames of a recorded microscopy video. Due to Brownian motion, the monomer particles appear as particle clouds, but they do not respond the applied AC electric field. **c** A microrobot is maneuvered along a race track adopted from a well-known video game. The image shows an overlay of successive frames of a recorded microscopy video and the racetrack.

microrobot approaching a cargo particle (shown in green in the image). As illustrated in the sketch above the microscopy image, fluid flows towards both particles in the direction of the electric field and is repelled perpendicularly to it. The microrobot and the cargo particle drift in the fluid flow caused by each other which results in an attractive interaction for the configuration shown.

When in direct contact, the microrobot sticks to the cargo via induced dipole-dipole forces (see Fig. 3b). Even though we apply an AC electric field, at any point in time the external field induces electric dipoles in both microrobot and cargo, which point in the same direction and thus result in a near-field attraction of the particles. This mechanism is well known and results in particle chain formation in crowded colloidal suspensions[7,29,30]. We then maneuvered the cargo-loaded microrobot around another monomeric particle as shown in the overlay image in Fig. 3b. For this image, we corrected the drift in the corresponding microscopy video by moving the tracked position of the monomeric non-cargo particle into the center of each frame.

Cargo release (shown in Fig. 3c) was achieved by switching off the electric field, changing the frequency from 250 Hz to 750 Hz, and then applying an AC electric field with an orientation roughly perpendicular to the previous field. The corresponding particle configuration and flow fields are illustrated in the sketch above the microscopy image,

where the fluid is repelled perpendicularly to the electric field from the particles' equators, resulting in their repulsion. Once the microrobot had moved sufficiently far from the cargo particle, the frequency was reset to 250 Hz. We found that increasing the frequency made it easier to execute cargo release, as the microrobots moved more slowly at higher frequencies. This is presumably caused by lower CPEO flow magnitudes. Additionally, we hypothesize that at higher frequencies the strength of the induced dipole-dipole force decreases relative to the force exerted by the fluid flow. It is worth to note that although the interaction between two parallel-aligned dipoles in this configuration is also repulsive, the particles will always return to a chain formation when they are subjected solely to dipole forces. Importantly, the field lines of the dipole-dipole forces start and end at the dipoles, which prevents the particles from escaping. A video of the cargo transport process is shown in the fourth part of Supplementary Movie 1.

**Controlled assembly of cargo particles into particle chains**
Using the same strategy for particle transport and release, we were also able to assemble several monomeric particles into a particle chain. As shown in Fig. 4 (cf. last part of Supplementary Movie 1), the microrobot can be controlled to sequentially pick up two individual cargo particles and drop them off in the vicinity of a third target particle. As a result of

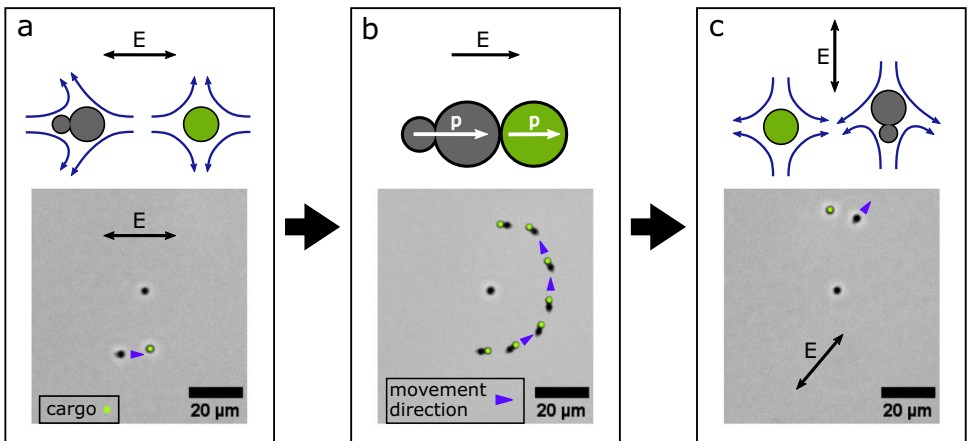

**Fig. 3 | Cargo pick-up, transport and release. a** A microrobot approaches a cargo particle. The direction of motion of the microrobot is indicated with a blue pointer and the cargo particle is labeled with a green dot. The orientation of the AC electric field is indicated with a black double arrow. The fluid flow arising around the microrobot and the cargo particle is illustrated with curved blue arrows in the sketch above and leads to an attraction. **b** A microrobot sticks to a cargo particle via induced dipole-dipole forces. Both are maneuvered around another monomeric particle. The instantaneous induced dipole moments are indicated with white arrows in the sketch above, the electric field is indicated with a black arrow. **c** A cargo particle is released from a microrobot by a quick change in direction of the external electric field. The fluid flow arising around the microrobot and the cargo particle now leads to repulsion.

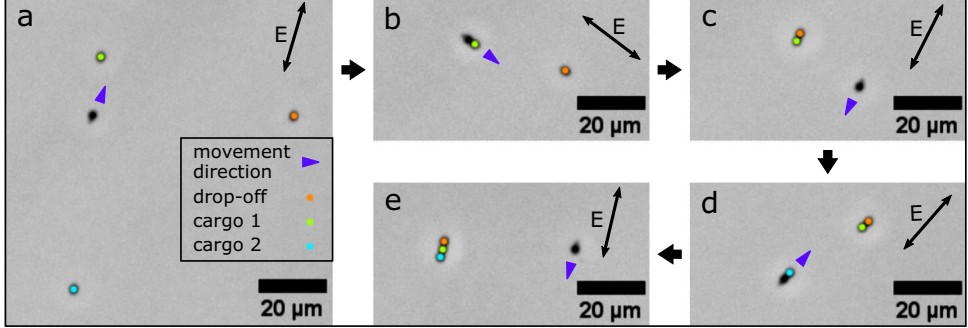

**Fig. 4 | Assembly of three monomeric particles into a particle chain. a** A microrobot approaches cargo particle 1 (labeled with a green dot). **b** The micro-robot, loaded with cargo 1, approaches another monomeric particle (orange), which serves as the drop-off location for cargo release. **c** The cargo particle sticks to the orange particle via induced dipole-dipole forces, while the microrobot is maneuvered towards cargo particle 2 (turquoise). **d** The microrobot loaded with cargo 2 heads back towards the drop-off location to release the cargo. **e** The two cargo particles and the orange target particle are assembled into a chain.

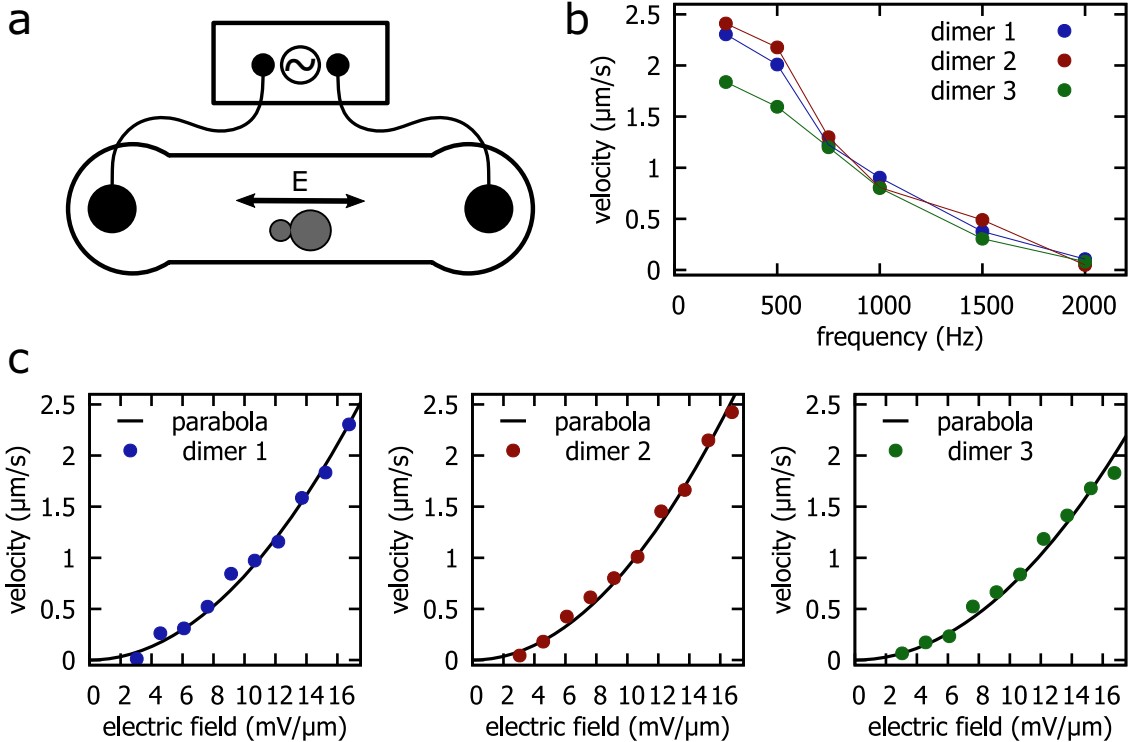

**Fig. 5 | Dependence of migration speed on frequency and amplitude of the AC field. a** Simplified measurement setup. An AC electric field is applied to a linear microscopy channel containing dimers. **b** Migration speed of three different dimers at a constant electric field amplitude of 16.8 mV/$\mu$m and varying frequency. The colored lines are a guide for the eye. **c** Migration speed of three different dimers at a constant frequency of 250 Hz and varying electric field amplitude. The black lines are parabolas ($v = \lambda E_0^2$) fitted to the velocity data. Source data are provided as a Source Data file.

the attractive induced dipole-dipole interactions between the monomeric particles, the three particles stick together and form a particle chain.

**Amplitude and frequency dependence of the CPEO mechanism**
Having established a novel approach for the manipulation of asymmetric colloidal dimers, we intended to verify whether the underlying propulsion mechanism indeed conformed with the theoretical framework for CPEO. We hypothesized that the migration speed of the dimers would scale similarly as the strength of the fluid flow around spherical monomer particles. Like ICEO, CPEO is a second-order phenomenon with respect to the applied electric field and thus the migration speed $v$ should scale with the electric field amplitude $E_0$ as $v \propto E_0^2$. However, the frequency dependence of CPEO is expected to differ substantially from that of ICEO. Notably, the fluid velocity around spherical particles caused by CPEO falls off to zero for frequencies exceeding the characteristic frequency $f_c = \frac{1}{2\pi} \frac{2D^+D^-}{R^2(D^++D^-)}$[5,7,31,32]. Here, $D^-$ and $D^+$ denote the diffusion coefficients of the buffer ions Cl$^-$ and Tris-H$^+$, which are $D^- = 2 \times 10^3 \, \mu\text{m}^2/\text{s}$ and $D^+ = 0.8 \times 10^3 \, \mu\text{m}^2/\text{s}$ at $T = 20\,°\text{C}$[33] and $R$ is the radius of the object. When applying the theory to our dimers, we can interpret $R$ as their typical size, which we take as the average radius of their constituent particles, and with the given parameters we obtain $f_c = 297$ Hz. $1/(2\pi f_c)$ corresponds to the time required for ions to diffuse over the distance $R$. By contrast, in the case of ICEO the characteristic frequency is derived

from the time required to charge the electric double layer on the particle, which is given by the RC time of the corresponding circuit[3,34].

We measured the migration speeds of three different dimers for several voltages and frequencies to verify the above hypothesis. To this end, we employed the setup shown in Fig. 5a where dimers are placed in a linear microchannel with two electrodes at its opposite inlets. As before, we prepared the dimers in 100 $\mu$M Tris-buffer, which we titrated to pH 8.4 by the addition of HCl and supplemented with 5.2 $\mu$M MgCl$_2$. We then recorded microscopy videos of the migration of the three dimers while applying AC fields with different frequencies and amplitudes. From the start and end positions of the dimers, we computed the distances covered and from these the average migration speeds. We also measured the speed of a monomeric particle as a reference (see Methods for details). Fig. 5c shows plots of the migration speeds versus the applied electric field amplitude at a constant frequency of 250 Hz. As shown, the experimental migration speeds are well described by a quadratic fit $v = \lambda E_0^2$. In Fig. 5b, the frequency dependence of the migration speeds of the three dimers is plotted for a constant electric field amplitude of 16.8 mV/$\mu$m. We find a decrease of the migration speed in the range of the characteristic frequency $f_c = 297$ Hz calculated for CPEO flows.

For comparison, we computed the characteristic frequency of ICEO flows around metal and uncharged dielectric spheres at our experimental conditions[3,34]. In addition, we calculated the expected slip velocities around spheres for CPEO and ICEO flows at the highest applied electric field amplitude 16.8 mV/$\mu$m (Table 1)[3,5,35]. The corresponding calculation can be found in the Supplementary Information. As mentioned, the frequency response of our dimers agrees best with the characteristic frequency predicted by CPEO, whereas $f_c$ predicted by ICEO for dielectric particles is two orders of magnitude off. The characteristic frequency predicted for strongly polarizable particles, such as metal particles, is closer to the experimentally observed value,

**Table 1 | Characteristic frequencies $f_c$ and velocity scales $v$**

|  | CPEO | ICEO metal | ICEO dielectric |
|---|---|---|---|
| Characteristic frequency | 297 Hz | 5.40 kHz | 98.0 kHz |
| Velocity scale | 17.5 $\mu$m/s | 175 $\mu$m/s | 0.33 $\mu$m/s |

but application of this model to our case is physically unreasonable as silica particles are not strongly polarizable. The value for the slip velocity around a sphere calculated from CPEO is found to be one order of magnitude larger than the observed migration speed of our dimers. This result is not unexpected since the slip velocity and the migration speed are not directly equivalent, as also demonstrated in the schematic diagram shown in Fig. 1c. The migration speed may be further reduced due to the additional drag caused by the nearby channel bottom. Importantly, the slip velocity calculated for ICEO flow around a dielectric sphere is approximately one order of magnitude lower than the observed dimer migration speed.

## Buffer dependence of the transport mechanism

We finally also characterized the buffer dependence of the microrobots' migration velocity. CPEO flows are caused by ion-selective surface conduction in the electric double layer at the particle surface, which depends on the zeta potential. The zeta potential, in turn, is a function of the surface charge of the particle and the ionic strength of the buffer solution. The flow is thus expected to be strongest for large surface potentials, i.e., under conditions with large surface charge densities and low ionic strengths. We recorded microscopy videos of dimers prepared in buffers with different concentrations of Tris, NaCl and NaOH, each supplemented with 5.2 $\mu$M MgCl$_2$. In addition, we tested Tris-buffer, NaCl and NaOH without any MgCl$_2$ and also a solution containing exclusively MgCl$_2$. For each buffer composition, we recorded tracks of at least five dimers, and we took care that every video contained at least one spherical particle as a reference. As before, we measured the migration speed by marking the start and end position of the dimer and dividing the resulting distance by the elapsed time. We also measured the migration speed of all spherical particles in each microscopy video and used it as a reference (see Methods for details). The results of these experiments are listed in the Supplementary Tables 1–8 and plotted in Fig. 6.

Overall, the velocities tended to decrease for increasing monovalent salt concentrations, approaching zero velocity for

concentrations around 1 mM, which is in the range expected for CPEO flows for typical values of the surface charge[7]. The details of the buffer dependence of the particle velocity are intricate, however. We found that supplementing the buffers with 5.2 $\mu$M MgCl$_2$ had a tremendous effect on the migration behavior. For Tris buffer, we found a strong enhancement of the velocity by MgCl$_2$. For NaOH, we found that dimers moved backwards in the absence of MgCl$_2$, while they moved forward in its presence. Notably, when using MgCl$_2$ in dH$_2$O only, we found backward movement that changed to forward movement at higher concentrations. When present alone and at low concentrations, either NaOH or MgCl$_2$ induced backward movement. However, when combined they induced forward movement. We found the largest migration velocities for NaOH and Tris buffer at pH 8.4, which we attribute to an increase in surface charge caused by the elevated pH. As even small amounts of MgCl$_2$ had an extreme effect on the migration velocity, we took special care to avoid any salt contamination in our samples (see Methods for details).

For an electrokinetic phenomenon such as CPEO, a complex dependence on buffer conditions is not unexpected. The reversal of the direction of movement of the particle dimers could be caused by either a complete flow inversion or by a more subtle change of the curvature of the flow lines around the particles depending on the ionic environment. At the microscopic level, these variations might be associated with the DNA molecules present on the colloidal surface, which are known interact strongly with Mg$^{2+}$ ions[36]. An extended discussion of this phenomenon is provided in the Supplementary Information, which is briefly summarized here.

In particular, we found that the theory by Fernández-Mateo et al.[5], which was developed for binary electrolytes with identical ion diffusion coefficients $D$, indeed predicts flow reversal around spherical particles for certain values of $D$. However, in this case flow reversal is restricted to scenarios with unrealistically high zeta potentials ($\zeta > 100$ mV) and rather low diffusion constants ($D \approx 500\ \mu$m$^2$/s) (Supplementary Figs. 3 and 4). We therefore experimentally examined the flow field around the microswimmers using tracer particles under

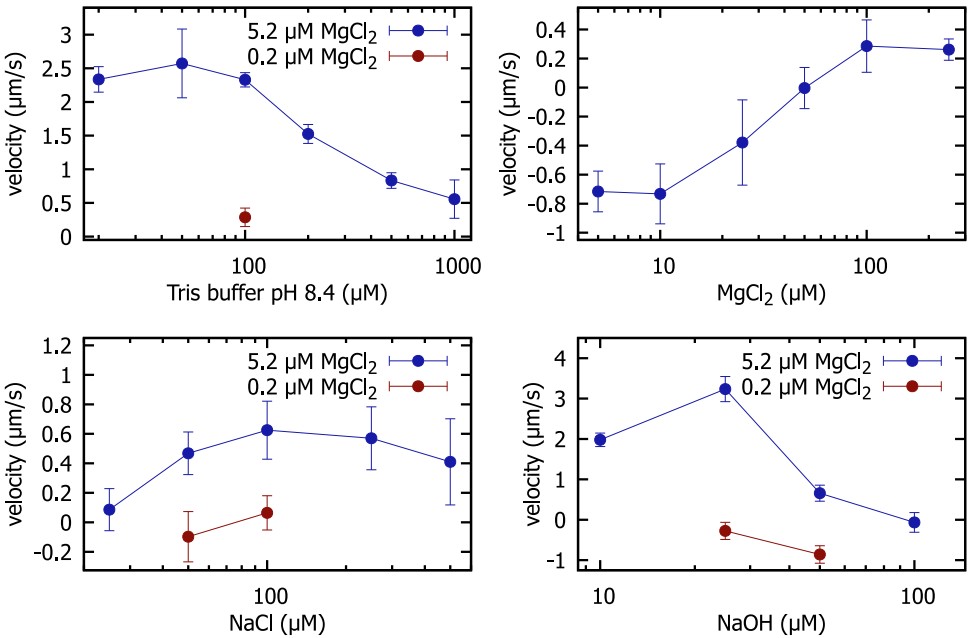

**Fig. 6 | Dimer velocities at a constant electric field amplitude of 16.8 mV/$\mu$m and constant frequency of 250 Hz for various buffer conditions.** Blue data points indicate measurements with MgCl$_2$ supplemented to the buffer, while red data points correspond to measurements without supplemented MgCl$_2$. Each data point is the average of at least 5 velocity measurements of different dimers, error bars indicate the corresponding standard deviations. Data points with negative velocities correspond to dimers moving backward, which we defined as a movement with the smaller particle in front. Source data are provided as a Source Data file.

**Fig. 7 | Selected frames from a video showing an individual yeast cell and a yeast cell doublet subjected to an AC electric field.** The cells migrate in opposite directions, parallel to the electric field, and pass each other.

buffer conditions leading to either backward (50 $\mu$M of NaOH and 0 $\mu$M of MgCl$_2$) or forward migration (25 $\mu$M of NaOH and 5 $\mu$M of MgCl$_2$) (Supplementary Movies 3 and 4). The results of these experiments suggest that the flow lines are differently curved around the swimmer particles for the two movement directions rather than simply inverted (Supplementary Fig. 5). This suggests that the flow magnitude around the smaller particle changes relative to that around the larger particle under different buffer conditions.

To determine whether the reversal of migration direction is linked to the DNA surface modification, we measured the mobility of our 1.0 $\mu$m silica particles with and without DNA using electrophoretic light scattering, under buffer conditions corresponding to either forward or backward migration. We observed no significant differences between both particle types, suggesting a negligible impact of the DNA. Interestingly, the presence of 5 $\mu$M MgCl$_2$ significantly altered the mobility of the monomers, which is consistent with the strong screening effect of the divalent Mg$^{2+}$ ions. We estimated the surface charge from the mobility using the Smoluchowski equation and the Poisson-Boltzmann equation, revealing that the additional 5 $\mu$M of MgCl$_2$ strongly altered the mobility without appreciably affecting the surface charge. Considering the pronounced effect of even small quantities of Mg$^{2+}$ ions on the mobility, we suggest that a detailed calculation of the dimensionless flow velocity for mixed electrolytes, including divalent ions, could provide mechanistic insight into the reversal of the migration direction of the microswimmers.

### Universality and interplay of CPEP and ICEP in AC Electrophoresis

As a general comment, we would like to note that AC electrophoresis (ACEP) in a homogenous AC-electric field is a generic phenomenon that should be expected for most particle types with a broken symmetry under the appropriate experimental conditions, i.e., at sufficiently low salt concentrations and an externally applied frequency that matches the characteristic frequency of the particles. The underlying mechanisms for ACEP are the well-established ICEP for metal particles and CPEP (described in this paper) for dielectric particles with a non-zero surface charge. Importantly, all electrostatically stabilized colloidal dispersions inherently meet the non-zero surface charge requirement for CPEO flows. In our previous work[7], we have demonstrated that a variety of particles, including fluorocarbon (FC) oil and lauric acid droplets, coacervates, silica particles, E. coli bacteria, and ground coffee, are subject to CPEO flows. As it is the case for our microrobots, CPEO flows around an arbitrary asymmetric particle will be asymmetric which will lead to a propulsion in a direction determined by the shape of the particle.

To confirm the universality of CPEP, we conducted experiments with a variety of charged dielectric particle types exhibiting broken symmetry, including fragmented ceramic, borosilicate glass, acrylic glass, agarose gel debris, ground coffee particles, and even yeast cells (Supplementary Movie 2). Using the setup illustrated in Fig. 5a, we exposed these particles to a homogeneous AC electric field with an amplitude of 16.8 mV/$\mu$m and a frequency of 250 Hz. As demonstrated in Supplementary Movie 2, we indeed observed the expected, directed

migration for all asymmetric particle types. The particles moved predominantly along the electric field lines, even though some migrated at an angle to the field. As an example, Fig. 7 shows three selected frames from a microscopy video, in which an individual yeast cell and a yeast cell doublet pass each other, while moving in opposite directions.

We conjecture that for composite particles (e.g., metallodielectric Janus particles), ICEP is not an isolated phenomenon. A CPEO flow may occur along with the ICEO flow generated on the metallic side, provided the dielectric Janus face is sufficiently charged. Given that ICEO and CPEO flows operate at different characteristic frequencies, their interplay would depend on the applied frequency. A similar argument can be made for particle dimers exposed to a vertical electric field. These should also be subject to CPEO flows in addition to the flows created at the electrode, when AC electric fields are applied within the frequency range defined by the corresponding characteristic frequency.

## Discussion
### Electrophoretic propulsion strategies

A wide range of AC-electrophoretic propulsion strategies have been developed in the past years, which differ from our approach in several aspects. In general, there are two possible electric field geometries - either microswimmers are positioned on top of large electrodes and exposed to vertical electric fields[9–15,37–41] or they are subjected to in-plane electric fields using remote electrodes[4,42] as employed in our study. In both configurations, the applied electric field must be large enough to generate a voltage drop across the particle on the order of the thermal voltage $k_BT/e$. Due to the shorter distance between the electrodes, vertical electric field setups typically require lower voltages, making them easier to implement.

Vertical electric field configurations, however, induce propulsion only in a random direction perpendicular to the field. To achieve directional motion or steerability with microswimmers in such setups, magnetic fields have been employed, requiring additional external magnets and the use of appropriate magnetic microparticles.[12,13,15,38,39] By contrast, employing in-plane electric fields - such as in this work - naturally provides alignment and steerability through electroorientation[18–21] of the microswimmers. This results in an overall simpler experimental setup and more design flexibility for the swimmers as they do not have to be magnetic.

The two electric field configurations can be implemented in combination with two basic microswimmer designs - metallo-dielectric Janus particles[4,10–12,14,15,37–42] and asymmetric colloidal dimers[9,13]. Both microswimmer designs are expected to exhibit propulsion in both electric field configurations. The propulsion mechanisms of each combination differ, each being associated with a distinct characteristic frequency that must be matched with the frequency of the external electric field in order to achieve propulsion. The characteristic frequencies range from the lower MHz range for Janus particles in vertical electric fields, which are driven by self-dielectrophoresis[11], to the lower kHz region for ICEO-driven metallo-dielectric Janus particles[4], to the 297 Hz needed to propel our asymmetric colloidal dimers by CPEO. Overall, the CPEO mechanism can be applied to the broadest class of

particles that can be used as microswimmers, as only particle asymmetry and a surface charge is required. This offers great design flexibility and even allows the use of biological and soft materials as microswimmers, overcoming the need for hard, durable substrate particles that are required for metal deposition to manufacture Janus particles.

To implement effective cargo transport, both attractive and repulsive interactions between microswimmer and cargo are necessary to establish an efficient loading and release mechanism. In our setup, this is accomplished through the geometry of the induced hydrodynamic flow, which can be either attractive or repulsive, depending on the particle-cargo configuration. The only alternative strategy known to balance these interactions employs Janus particles in vertical electric fields, where cargo particles are either repelled or attracted to the dielectric or metallic side based on the applied frequencies. This strategy has been previously implemented using specially designed ferromagnetic metallo-dielectric Janus particles, allowing for steerability and directed cargo transport[12,15,39].

In conclusion, we have introduced a novel approach towards AC electrophoretic (ACEP) manipulation of colloidal microswimmers, namely concentration-polarization electrophoresis (CPEP), which facilitates precise electrical control over two-dimensional movements. In contrast to other approaches for electrically driven swimmers, the utilization of concentration-polarization electroosmosis (CPEO) and electro-orientation enables the use of in-plane electric fields to move the particles in the direction of the field lines, rather than perpendicular to them, as in other approaches. Directed movement requires asymmetric particles with a surface charge, but the particles themselves do not need to be 'Janus' or magnetic, which broadens the design possibilities for electrically controlled microrobots. In our case, two differently-sized silica particles were connected using DNA linker molecules. We employed a relatively simple setup to achieve 2D actuation, which did not require additional magnetic fields, as in the case of dimers or Janus particles subject to a vertical electric field, nor computer-controlled feedback, as in the case of dielectrophoretically driven microswimmers. As demonstrated by the joystick-controlled actions, our approach is of immediate interest for applications in microrobotics. The microrobots can move along arbitrarily chosen paths in 2D and can be directed to controllably pick up, release, and also assemble particles into groups. Further, we confirmed that the dependence of our microrobots' migration speed on the AC electric-field frequency, amplitude, and electrolyte concentration aligns with the theoretical expectations for CPEP. Finally, we confirmed that CPEP applies to a broad class of dielectric particles with a broken symmetry and non-zero surface charge by observing the directed migration of a variety of asymmetric particles subject to a homogenous AC-electric field. From this, we conclude that AC electrophoresis (ACEP) in a homogeneous AC-electric field, governed by ICEP for metal particles and CPEP for dielectric particles, is a universal phenomenon expected for most asymmetric particles.

Looking ahead, the generic nature of ACEP opens up opportunities for frequency-dependent particle sorting or precise positioning of particles, as their unique size, shape, and composition should result in distinct frequency responses, enabling their selective manipulation. Further, it is conceivable to let microrobots assemble other microparticles into defined superstructures, which themselves could then also act as microrobots, potentially laying the basis for a simple form of 'self-replication'. One of the main challenges for future applications, however, will be the realization of operating conditions that are compatible with useful chemical or biochemical reactions.

## Methods

### Functionalization and dimerization of colloidal particles

Carboxylated silica spheres with diameters 1.01 $\mu$m (Lot: SiO$_2$-COOH-AR756-5ml) and 2.12 $\mu$m (Lot: SiO$_2$-COOH-AR1060-5ml) were purchased from microParticles GmbH. We modified the surface of the silica spheres by activating the carboxyl groups with 1-Ethyl-3-(3-dimethyl-aminopropyl) carbodiimide (EDC) and coupling them to amino-modified DNA[43,44]. The colloids were reacted in 200 $\mu$L of 100 mM MES buffer (pH 4.8 adjusted with HCl and NaOH) containing 250 $\mu$M amino-modified DNA and 250 mM EDC (Merck: Art. No. E6383-1G) on a rotator at room temperature for 3 h. We used colloid concentrations of $11.35 \cdot 10^9$/mL and $50 \cdot 10^9$/mL of the 2.12 $\mu$m colloids and the 1.01 $\mu$m colloids, respectively, to account for the different surface areas of the colloids. The colloids were then washed and incubated extensively in borate buffer (boric acid adjusted to pH 8.2 with NaOH) and deionized water to get rid of remaining reaction components and to hydrolyze unreacted activated carboxyl groups. We avoided using buffers containing amino groups for washing as we wanted to preserve the negative surface charge of the colloids. An extended protocol with details on the washing procedure is given in the Supplementary Materials. Finally, the colloids were diluted to concentrations of $2.27 \cdot 10^9$/mL (2.12 $\mu$m colloids) and $10 \cdot 10^9$/mL (1.01 $\mu$m colloids) in deionized water, shock frozen in liquid nitrogen and stored at -80 °C.

Our two DNA strands are 60 nucleotides (nt) long and are each composed of a 30 nt long spacer region followed by a 30 nt region which is complementary to the corresponding region on the other strand. The spacer provides flexibility in the distance between the colloids where hybridization can take place. We designed our DNA sequences with NUPACK[45] such that they have no secondary structure. The oligonucleotides were purchased from Integrated DNA Technologies as dried pellets, their sequences are listed in the Supplementary Information. We diluted our DNA strands in deionized water and stored them at −20 °C.

### Microrobot assembly

We assembled our microrobots by incubating concentrations of approximately $1.6 \cdot 10^9$/mL of each colloid with 4 mM MgCl$_2$ in a reaction volume of 25 $\mu$L for 45 min on a rotator. The above colloid concentration assumes that no colloids were lost in the above washing procedure. The reaction is stopped by rapid dilution of the sample by a factor of 1 to 1000 in deionized water. The sample is handled with special care as we found that shaking causes the microrobots to disintegrate. For further use, we usually diluted our microrobots again by a factor of 1 to 20 in a buffer of choice. The microrobots were assembled freshly for every day of experiments.

### Sample preparation

In initial experiments, we found a reduction in the migration speed after washing our pipette tips. We therefore suspected that the pipette tips contained trace amounts of divalent ions. In order to establish stable and reproducible behavior of the swimmers, we henceforth cleaned all pipette tips and the sample chamber with deionized water before usage. With every fresh pipette tip, we pipetted deionized water three times before pipetting an actual sample. We also washed all used reaction tubes with deionized water before usage, vortexed them and removed the deionized water again. For our screening experiments, we used commercial microscopy chambers purchased from ibidi ($\mu$-Slide VI 0.4; Cat.No:80601). Before usage, the microscopy chambers were filled three times with deionized water and then blown dry with nitrogen gas. For our experiments with Tris-buffer, we created a 50 mM stock solution at pH 8.4 by titrating Tris (Carl Roth: Art. No. 4855.2) with HCl. We avoided using NaOH in case of overshooting pH 8.4 as this would have resulted in an unknown concentration of additional NaCl in the buffer.

### Preparation of fragmented particles

We produced fragmented particles from a variety of materials as described below. Following their production, the fragments were washed to remove electrolytes and prepare them for use in our

experiments. To this end, we immersed the particles in 1 mL of deionized water in a 1.5 mL Eppendorf tube. We centrifuged the resulting suspension, adjusting the duration and centrifugal forces depending on the stability of the particles. The supernatant was subsequently discarded, and the tube refilled with 1 mL of deionized water. This washing process was repeated three times. Finally, we diluted the washed suspension by an empirical factor using the same buffer as in our microrobot experiments, i.e., 100 $\mu$L Tris buffer at pH 8.4, supplemented with 5.2 $\mu$L MgCl$_2$. We finally adjusted the dilution to obtain a particle density appropriate for imaging. *Agarose gel:* We mixed 1.5 g of agarose powder (Agarose NEEOP Ultra-Qualität; Art.-NR.2267.3) from CARL ROTH with 30 mL of deionized water. We dissolved the agarose by heating in a microwave oven. We then poured the hot agarose solution into a glass Petri dish and let it cool and solidify for approximately 20 min. Once solidified, we gently scratched the agarose surface using a scalpel, moving the blade perpendicularly to the cutting edge across the gel surface. We collected around 20 mg of the agarose gel fragments in a 1.5 mL Eppendorf tube and filled it up to 1 mL with deionized water. For the washing process, we centrifuged for 1 min at 1000 rcf. Finally, we diluted the washed gel fragments by a factor of 20. *Borosilicate glass:* To obtain glass fragments, we mechanically pulverized a borosilicate glass capillary (BOROSILICATE GLASS; ITEM #: BF150-86-7.5) from Science Products GmbH placed between two glass microscopy slides (Objektträger 76 x 26 mm; Art.Nr.0656) from CARL ROTH. We recovered the resulting glass powder by pipetting 100 $\mu$L dH$_2$O onto the slide and transferred it into a 1.5 mL Eppendorf tube that was filled up to 1 mL with deionized water. For the washing process, we centrifuged for 1 min at 1000 rcf, followed by dilution 1:5. *Acrylic glass:* Acrylic glass fragments (Poly(methyl methacrylate)) were produced using a fine metal file. We collected approximately 2 mg of the resulting powder suspended it in 1 mL of deionized water within a 1.5 mL Eppendorf tube. For the washing process, we centrifuged for 1 min at 1000 rcf, followed by dilution 1:10. *Brewers yeast:* We extracted a turbid suspension of brewer's yeast cells from a sample of Bavarian wheat beer with a pipette and transferred 1 mL of the suspension into a 1.5 mL Eppendorf tube. For the washing process, we centrifuged at 250 rcf for 4 min. Finally, we diluted the washed cells by a factor of 20 in Tris buffer. *Ceramic:* We created small ceramic fragments by placing a splinter from a coffee mug of the size of a fingertip between two aluminum sheets and grinding it. We transferred approximately 2 mg of the resulting powder into a 1.5 mL Eppendorf tube and filled it with 1 mL dH$_2$O. For washing, we centrifuged at 1000 rcf for 1 min, and finally diluted the washed ceramic fragments 1:5 in Tris buffer. *Coffee:* We pipetted 1 mL of coffee (prepared using standard procedures using very finely ground coffee beans) into a 1.5 mL Eppendorf tube. For the washing process, we centrifuged the suspension at 1000 rcf for 1 min, followed by dilution 1:5.

### Video editing
We used ImageJ to edit our videos. We corrected the drift in the corresponding microscopy video by tracking the monomeric particles with the ImageJ plugin TrackMate[46] and shifting the recorded video by their displacement. Overlay images were created by computing the minimum intensity of a collection of frames from a microscopy video. We adjusted the contrast and brightness of our videos and images such that they appear alike. The final video editing was done with the freely available software Shotcut[47].

### Data analysis and data availability
We marked the start and end positions of every microrobot and reference particle in a video and saved the coordinates with the corresponding frame number. We also recorded the instantaneous orientations of every microrobot, which lets us identify backward and forward movements. For that purpose, we used an ImageJ macro to automatize the data analysis. We computed the velocities of all microrobots and reference particles by subtracting the y-coordinates

of the start and end positions and dividing the result by the elapsed time. The elapsed time was extracted from the metadata of the corresponding video. We then computed the average velocity of all reference particles in a video and subtracted the velocity of every microrobot in a video by the result, which gives us the corrected microrobot velocities. The average and standard deviation are then computed from the corrected velocities of all videos with the same buffer conditions. All measured velocities and the recorded microrobot orientations for our buffer characterization experiments are listed in the Supplementary Tables 1–8. In our frequency and electric field characterization experiments, we were interested in the response of a single microrobot and reference particle. We therefore applied a simplified data analysis procedure and measured only the speed of the single microrobot and reference particle in a video. The recorded measurements are listed in the Supplementary Tables 9–11. All tables are also provided as source data files in the Excel format.

### Setup design and operation
The design of our self-made sample chamber is inspired by that of Kopperger et al.[23]. Our sample chambers were milled from of a 5 mm thick PMMA sheet using a micro-milling machine. The bottom part of our microscopy chamber is a glass cover slide (Carl Roth: Art. No. CEX2.1). The PMMA part is glued to the glass slide with Dichloromethane (Carl Roth: Art. No. 8424.2). We use platinum wires (Merck: Art. No. 267201-400MG) with a diameter of 0.5 mm as electrodes. The electrode mounting is milled from a 10 mm PMMA sheet. We use a standard Xbox Controller (PDP 049-012-EU-RD Controller Xbox Series X Rot) connected via an USB-cable to the microscope computer to control electrical signals generated by the sound card of the computer. A more detailed description, including photographs of the setup, can be found in the Supplementary Materials. For microscopy, we used an Olympus IX71 inverted microscope equipped with a 20x objective (Olympus UPlanFL N 20x/0.50) and an 100x objective (Olympus PlanApo 100x/1.40 Oil). During our screening experiments we monitored the current and voltage with a digital oscilloscope (PicoScope 2000) to avoid systematic errors. Electric signals for our screening experiments were created with a function generator (RIGOL DG812) and amplified with an amplifier built in-house.

### Programming
We used the online tool ChatGPT[48] based on GPT3[49], a Generative Pretrained Transformer developed by OpenAI, to assist with programming.

## Data availability
Source data are provided as a Source Data file, comprising Excel files that enumerate all individual measurements utilized in the graphs depicted in Figs. 5 and 6, alongside the raw zeta sizer data employed in the bar graph in Supplementary Fig. 6. The microscopy video data generated in this study have been deposited in the Dryad database under accession code DOI: 10.5061/dryad.9w0vt4bn7. Source data are provided with this paper.

## Code availability
The program code for the control of our microrobots and the data analysis scripts are available by the authors upon request.

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

## Acknowledgements

We thank Thomas Mayer for helping us with the DNA strand design. We thank Jonathan List for providing the second stage of our custom-built amplifiers. This work was funded by the Deutsche Forschungsgemeinschaft (DFG, German Research Foundation) Project-ID 364653263 TRR 235. We acknowledge additional support via the Excellence Strategy of the Federal Government and the Länder through the TUM Innovation Network Robotic Intelligence in the Synthesis of Life (RISE).

## Author contributions

F.K. and F.C.S. conceived the project. F.K. planned the experiments, built the experimental setup, performed all experiments, and analyzed the data. F.K. wrote the paper with support by F.C.S.

## Funding

## Competing interests

The authors declare no competing interests.
