## [Peer Review File · Nature Communications]

REVIEWER COMMENTS

Reviewer #1 (Remarks to the Author):

This paper studies the propulsion of asymmetric particle dimers connected with DNA linkers, that are steered controlling an AC field. The mechanism for propulsion is claimed to be novel, through concentration polarization electroosmosis (instead of the classical ICEO mechanism). The authors demonstrate impressive control of the particle dimers, steering them with a joystick and demonstrating that it is possible to make the dimers pick up and drop single particles. The mechanism is on the surface simpler than other possibilities, allowing in plane electric fields with controllable orientation to orient and propel the particles and not requiring the use of a magnetic field. The majority of the paper outlines the mechanism and demonstrates that it works through movies of particles in a specially designed microfluidic device. The last part of the paper carries out a number of experiments and analyses designed to show that the propulsion mechanism in the system is CPEO instead of ICEO.

Overall I find this to be a careful study, which is of potential interest to the community. To be convinced of its suitability for this journal, I'd like the authors to explain to me the effectiveness of this propulsion mechanism vis a vis the other mechanisms that have been presented in the literature. There is a rich literature -- and every paper is like this one, explaining how its mechanism is important, but never comparing in any depth to the others. This paper goes to lengths to distinguish its mechanism from ICEO, and this seems convincing. But this is different than explaining ****why**** it is either useful or scientifically interesting to have a propulsion mechanism with the properties that this one does. At a high level, what do we learn from this study? How will it change science or technology? The paper is silent on these questions. I'd like to understand why I should care about the results. The authors seem to be claiming that the mechanism allows flexibility with eg picking up particles and putting them down; can other mechanisms do this? Or is what is different here the particular frequency dependence? If it is the latter, then which frequency dependence is more "useful"? Broad context is not given here to put the core results in the context of the broader literature.

Reviewer #2 (Remarks to the Author):

Comments on the manuscript "Microrobots Powered by Concentration Polarization Electrophoresis (CPEP)" by Florian Katzmeier and Friedrich C. Simmel

The authors present the experimental realization of microscopic objects that can be moved by homogeneous AC electric fields along arbitrarily chosen paths within a 2D plane. The microscopic objects are asymmetric particle dimers fabricated by joining two silica spheres of different radii (1 and 2 μm diameters) using DNA linker molecules. The AC electric fields create steady electrokinetic flow around the colloidal particles caused by concentration polarization electro-osmosis (CPEO). Because of the broken fore-aft symmetry, the flow results in a phoretic motion of the asymmetric particle dimer. The authors control the orientation and motion of the dimers with the electric field generated by four electrodes, which create electric fields with any orientation within the 2D plane. These dimers are shown to pick up monomeric particles and release them at desired positions.

I think that the experiments have been performed with care, they are novel, with potential interesting applications, and with some advantages versus other approaches of microrobot particles. The dependence of dimer velocity with frequency, and electric field amplitude points to CPEO flows around the particles. The disappearance of the motion at high ionic strength (beyond 1mM) is also an indication of electrokinetic flow. However, the dimer speed versus concentration of MgCl_2 is not clearly correlated with CPEO flow. For this reason, prior to be published, I think that the following experiments should be carried out:

A) It would be interesting to have the experimental values of monomer mobilities as a function of MgCl_2 concentration. This could tell us whether the presence of MgCl_2 is increasing the surface charge of the particle or not, leading to an increase of electrokinetic flow.

B) In addition, another important experiment is to observe the experimental streamlines created by the dimers in cases where it moves in the direction defined by the big particle and in the opposite direction. These observations could be done with fluorescent tracers of submicron size, and will confirm that the propulsion is generated by some kind of flow like in figure 1c.

Other comments:

- Page 5. The alignment of the dimers could also be due to some hydrodynamic flow in combination with the dipole torque.

- Page 8. Cargo release is explained by assuming that at higher frequencies the induced dipole-dipole force decreases in strength relative to the force exerted by the fluid flow. However, if the electric field is applied perpendicularly, dipole-dipole interaction is also repulsive. In principle, there is no reason to assume that dipole strength decreases with frequency, and experimentally what is observed is the reduction of CPEO flow with frequency.

- Page 13 and supplementary information. Although there is fluid flow reversal by changing the diffusion coefficients even for symmetric electrolytes with equal diffusivities, the values needed for

this are one order of magnitude lower than the experimental ones. I think that this should be said in the main text.

Katzmeier & Simmel - Reply to the Reviewer comments:

The manuscript has been thoroughly revised in response to the reviewer comments, and additional experiments were performed. Changes to the manuscript and new text sections are highlighted in the revised manuscript in green.

Reviewer #1 (Remarks to the Author):

This paper studies the propulsion of asymmetric particle dimers connected with DNA linkers, that are steered controlling an AC field. The mechanism for propulsion is claimed to be novel, through concentration polarization electroosmosis (instead of the classical ICEO mechanism). The authors demonstrate impressive control of the particle dimers, steering them with a joystick and demonstrating that it is possible to make the dimers pick up and drop single particles. The mechanism is on the surface simpler than other possibilities, allowing in plane electric fields with controllable orientation to orient and propel the particles and not requiring the use of a magnetic field. The majority of the paper outlines the mechanism and demonstrates that it works through movies of particles in a specially designed microfluidic device. The last part of the paper carries out a number of experiments and analyses designed to show that the propulsion mechanism in the system is CPEO instead of ICEO.

Overall I find this to be a careful study, which is of potential interest to the community.

Reply: We would like to thank the reviewer for the favorable and supportive evaluation.

To be convinced of its suitability for this journal, I'd like the authors to explain to me the effectiveness of this propulsion mechanism vis a vis the other mechanisms that have been presented in the literature. There is a rich literature -- and every paper is like this one, explaining how its mechanism is important, but never comparing in any depth to the others. This paper goes to lengths to distinguish its mechanism from ICEO, and this seems convincing. But this is different than explaining **why** it is either useful or scientifically interesting to have a propulsion mechanism with the properties that this one does. At a high level, what do we learn from this study? How will it change science or technology? The paper is silent on these questions. I'd like to understand why I should care about the results. The authors seem to be claiming that the mechanism allows flexibility with eg picking up particles and putting them down; can other mechanisms do this? Or is what is different here the particular frequency dependence? If it is the latter, then which frequency dependence is more "useful"? Broad context is not given here to put the core results in the context of the broader literature.

Reply: We agree with the reviewer that the literature on microswimmers is vast and it is difficult to identify the significance of our particular work. There are several important aspects, which we tried to explain in the revised manuscript in more detail. First, our approach works with electric fields oriented in the plane, enabling better control than available for most other microswimmers (using perpendicular fields and/or magnetic fields for assistance). Second, we demonstrate that the mechanism is generic to all asymmetric dielectric particles carrying a surface charge. We performed additional experiments with a wide range of particles, including yeast cells, that show that they can be easily manipulated like our "designed" microswimmers. Third, the CPEP effect (for dielectric particles) is subsumed together with the previously described ICEP effect (for metallic particles) as two examples of "AC electrophoresis", suggesting a step towards a general framework for AC manipulation of colloidal particles. We believe that this combination -- ease of implementation and ability to control microswimmers, the generic nature of the phenomenon, and also the progress in theoretical understanding -- makes this work significant.

Reviewer #2 (Remarks to the Author):

Comments on the manuscript "Microrobots Powered by Concentration Polarization Electrophoresis (CPEP)" by Florian Katzmeier and Friedrich C. Simmel

The authors present the experimental realization of microscopic objects that can be moved by homogeneous AC electric fields along arbitrarily chosen paths within a 2D plane. The microscopic objects are asymmetric particle dimers fabricated by joining two silica spheres of different radii (1 and 2 μm diameters) using DNA linker molecules. The AC electric fields create steady electrokinetic flow around the colloidal particles caused by concentration polarization electro-osmosis (CPEO). Because of the broken fore-aft symmetry, the flow results in a phoretic motion of the asymmetric particle dimer. The authors control the orientation and motion of the dimers with the electric field generated by four electrodes, which create electric fields with any orientation within the 2D plane. These dimers are shown to pick up monomeric particles and release them at desired positions.

I think that the experiments have been performed with care, they are novel, with potential interesting applications, and with some advantages versus other approaches of microrobot particles.

Reply: We thank the reviewer for this positive assessment.

The dependence of dimer velocity with frequency, and electric field amplitude points to CPEO flows around the particles. The disappearance of the motion at high ionic strength (beyond 1mM) is also an indication of electrokinetic flow. However, the dimer speed versus concentration of MgCl_2 is not clearly correlated with CPEO flow. For this reason, prior to be published, I think that the following experiments should be carried out:

A) It would be interesting to have the experimental values of monomer mobilities as a function of MgCl_2 concentration. This could tell us whether the presence of MgCl_2 is increasing the surface charge of the particle or not, leading to an increase of electrokinetic flow.

Reply: We thank the reviewer for this suggestion. We carried out the suggested experiment and found that MgCl_2 indeed has a strong influence on the mobility of the particles (in fact independent of the DNA functionalization), which we attribute to the stronger screening effect of Mg^{2+} compared to monovalent ions. An extended discussion of this observation is now added to the main text and the SI.

B) In addition, another important experiment is to observe the experimental streamlines created by the dimers in cases where it moves in the direction defined by the big particle and in the opposite direction. These observations could be done with fluorescent tracers of submicron size, and will confirm that the propulsion is generated by some kind of flow like in figure 1c.

Reply: This is a very reasonable suggestion – the corresponding experiments turned out to be technically quite challenging as we needed to track moving particles and smaller tracer particles surrounding them to be able to map the flow field around them. In particular for the (slowly) backward moving particles this was difficult. The experiments are now described in the Supplementary Information. The flow fields agree with our initial expectation depicted schematically in Fig. 1. Furthermore, the results indicate that the reversal of movement direction for certain buffer conditions is caused by a change in the shape (curvature) of the streamlines around the particles, and not by a "simple" reversal of the EO flow.

Other comments:

- Page 5. The alignment of the dimers could also be due to some hydrodynamic flow in combination with the dipole torque.

Reply: We agree that probably both effects play a role and now explicitly mention this point in the revised manuscript.

- Page 8. Cargo release is explained by assuming that at higher frequencies the induced dipole-dipole force decreases in strength relative to the force exerted by the fluid flow. However, if the electric field is applied perpendicularly, dipole-dipole interaction is also repulsive. In principle, there is no reason to assume that dipole strength decreases with frequency, and experimentally what is observed is the reduction of CPEO flow with frequency.

Reply: We agree that there is no direct evidence that the dipole force decreases in strength relative to the fluid flow. We now write in conclusion that the main reason for the easier cargo release is mainly due to the overall lower velocities and fluid flow strengths at higher frequencies, which give the operator more time to precisely execute the joystick operations required for the cargo release. However, we still point out that dipole-dipole forces alone do not permit the escape of the cargo particle, since dipole-dipole forces are overall attractive.

- Page 13 and supplementary information. Although there is fluid flow reversal by changing the diffusion coefficients even for symmetric electrolytes with equal diffusivities, the values needed for this are one order of magnitude lower than the experimental ones. I think that this should be said in the main text.

Reply: We agree and mention this point explicitly in the revised discussion and SI. Our experiments showing the strong impact of Mg^{2+} and also the experimentally observed change in the curvature of the streamlines points towards a more complex underlying mechanism that is not captured by the simple "equal diffusivity" model.

REVIEWERS' COMMENTS

Reviewer #1 (Remarks to the Author):

I appreciate the author's clarity in comparing their mechanisms to others. I must admit that it still is hard for me to judge the significance of this mechanism relative to others -- though acknowledge that there is a broad community interested in swimming mechanisms, and the authors have now differentiated the advantage of theirs relative to others. I especially appreciate their demonstration that the mechanism works for a range of different objects.

Reviewer #2 (Remarks to the Author):

Second revision on "Microrobots Powered by Concentration Polarization Electrophoresis (CPEP)" by Florian Katzmeier and Friedrich C. Simmel.

I am very pleased with the responses of the authors to my previous comments and the realization of new experiments. In my opinion, the new experiments indicate the following.

- With respect to the presence of $MgCl_2$: this reduces the mobility of the monomers, which would not explain the notable increase of the CPEP of the dimers using the current theory. I agree with the authors that it is necessary to go beyond the theory for ions with equal diffusivities.

- With respect to the observations of flows around the dimers: These show that the change in particle direction is not due to a "simple" reversal of the flow, but to a change in the shape (curvature) of the streamlines around the particles. In my opinion, the latter could be consistent with a change in the relative intensity of the flow around the small particle becoming stronger than the flow around the big one.

As I said in my first comments, I think that the CPEP experiments are novel, with potential interesting applications, and with some advantages versus other approaches of microrobot particles. I recommend publication.

Reviewer #1 (Remarks to the Author):

I appreciate the author's clarity in comparing their mechanisms to others. I must admit that it still is hard for me to judge the significance of this mechanism relative to others -- though acknowledge that there is a broad community interested in swimming mechanisms, and the authors have now differentiated the advantage of theirs relative to others. I especially appreciate their demonstration that the mechanism works for a range of different objects.

Reply: We thank the reviewer for the very positive assessment.

Reviewer #2 (Remarks to the Author):

Second revision on "Microrobots Powered by Concentration Polarization Electrophoresis (CPEP)" by Florian Katzmeier and Friedrich C. Simmel.

I am very pleased with the responses of the authors to my previous comments and the realization of new experiments. In my opinion, the new experiments indicate the following.

- With respect to the presence of $MgCl_2$: this reduces the mobility of the monomers, which would not explain the notable increase of the CPEP of the dimers using the current theory. I agree with the authors that it is necessary to go beyond the theory for ions with equal diffusivities.

Reply: We appreciate the reviewer's agreement with our conclusion.

- With respect to the observations of flows around the dimers: These show that the change in particle direction is not due to a "simple" reversal of the flow, but to a change in the shape (curvature) of the streamlines around the particles. In my opinion, the latter could be consistent with a change in the relative intensity of the flow around the small particle becoming stronger than the flow around the big one.

Reply: We agree with the reviewer that it could indeed be consistent with a change in the relative intensity of the flow around each particle. We have added this conclusion to the main text.

As I said in my first comments, I think that the CPEP experiments are novel, with potential interesting applications, and with some advantages versus other approaches of microrobot particles. I recommend publication.

Reply: We thank the reviewer for the very positive assessment.